# The Distribution of DOM in the Wanggang River Flowing into the East China Sea

**DOI:** 10.3390/ijerph19159219

**Published:** 2022-07-28

**Authors:** Jie Ma, Dongyan Pei, Xuhan Zhang, Qiuying Lai, Fei He, Chao Fu, Jianhui Liu, Weixin Li

**Affiliations:** 1Nanjing Institute of Environment Sciences, Ministry of Ecology and Environment, Nanjing 210023, China; majie@nies.org (J.M.); laiqiuying@nies.org (Q.L.); 2School of Environmental Science and Engineering, Nanjing University of Information Science and Technology, Nanjing 210044, China; 20201248123@nuist.edu.cn; 3College of Hydrology and Water Resources, Hohai University, Nanjing 210024, China; 201601010126@hhu.edu.cn (X.Z.); 201605010005@hhu.edu.cn (C.F.); 201601010030@hhu.edu.cn (J.L.)

**Keywords:** DOM, river, three-dimensional fluorescence spectral

## Abstract

Dissolved organic matter (DOM) is a central component in the biogeochemical cycles of marine and terrestrial carbon pools, and its structural features greatly impact the function and behavior of ecosystems. In this study, the Wanggang River, which is a seagoing river that passes through Yancheng City, was selected as the research object. Three-dimensional (3D) fluorescence spectral data and UV–visible spectral data were used for component identification and source analysis of DOM based on the PARAFAC model. The results showed that the DOM content of the Wanggang River during the dry season was significantly higher than during the wet season; the DOM content increased gradually from the upper to lower reaches; the proportion of terrigenous components was higher during the wet season than during the dry. UV–Vis spectral data a_280_ and a_355_ indicated that the relative concentrations of protein-like components in the DOM of the Wanggang River were higher than those of humic-like components, and the ratio of aromatic substances in the DOM of the Wanggang River water was higher during the wet season. The DOM in the Wanggang River was dominated by protein-like components (>60%), and the protein-like components were dominated by tryptophan proteins (>40%). This study showed that the temporal and spatial distributions of DOM in rivers can be accurately determined using 3D fluorescence spectroscopy combined with the PARAFAC model. This provides useful insight into the biogeochemical process of DOM in rivers of coastal areas.

## 1. Introduction

Dissolved organic matter (DOM) is made up of compounds with complex compositions and structures [1,2] and is an important chemical component of natural water systems [3]. DOM includes humic acids (HA), fulvic acids (FA) [4], and various hydrophilic organic acids, carboxylic acids, amino acids, carbohydrates, etc. [5,6,7]. The molecular structures, compositional content, aromaticity, and degree of humification of DOM affect the migration and transformation of organic pollutants and heavy metals [8,9]. As the connection between the two largest carbon pools, marine and terrestrial [10,11,12], rivers are the main transportation pathway of terrigenous substances to the ocean and act as important links in the global carbon cycle [13,14,15]. Therefore, information about the dynamic changes of DOM will not only help to understand the carbon cycling processes of river ecosystems but also improve the understanding of ecological and environmental effects of terrigenous organic matter.

The composition of river DOM and its downstream/seaward flux are mainly controlled by land use types in the river basin, hydrological processes, and the transformation and transportation of organics within the river. Recently, land use patterns in watersheds have undergone significant changes due to human activities [16]. These activities include the destruction of forest vegetation, changes in agricultural farming methods, urbanization, construction of reservoirs and dams, and the inflow of large amounts of industrial and agricultural discharge, residential wastewater, and aquaculture wastewater into rivers [17,18,19,20]. All of these direct activities affect the availability, transportation, release, and retention of DOM in rivers [21,22,23]. It remains unclear how the seasonal spatial variation will influence the characteristics of various composed DOM in the complex seaward river ecosystem.

The fluorescence spectra are based on the three-dimensional excitation-emission matrix (EEM), which has been verified to be practical for tracing DOM sources in aquatic environments [24]. Common indicators that can be retrieved from fluorescence spectra include the biological index (BIX), fluorescence index (FI), and the relative ratio of fluorescent components, which provide valuable information on the structure and composition of organic matter [25,26,27]. FI and BIX characterize terrigenous organic and autogenic sources and newly generated organics, respectively [6,28]. Recently, the PARAFAC model has been further analyzed by emission fluorescence spectroscopy and spectroscopic indices to characterize DOM composition based on the 3D fluorescence spectra.

The Wanggang River is a key seagoing river that passes through Yancheng City. It has various tributaries making up a large and dense river network. The land bordering the river is used for a variety of purposes and is a rich and complex pollutant source.

In this study, 3D fluorescence spectra and the PARAFAC method were used, in combination with UV–Vis spectra, to investigate the composition of DOM in the water of the Wanggang River during the wet and dry seasons of 2021. The two hypotheses in this study are as follows: (1) DOM fluorescence intensities gradually increase from the upper reaches to the lower reaches of the Wanggang River; (2) DOM concentrations in the Wanggang River decrease during the wet season.

## 2. Materials and Methods

### 2.1. Sample Collection and Pre-Processing

A total of 30 sampling sites were established according to their topographical features, hydrological characteristics, distribution of tributaries, and typical pollution types (key agricultural non-point sources, industrial agglomeration outlets, and urban centers) in the Wanggang River Basin. Sample collection was carried out in May 2021 (the average water temperature reached 20 °C) and September 2021 (the average water temperature reached 32 °C). 2 L water samples were collected with a plexiglass sampler, stored in brown high-density polyethylene bottles, and brought back to the laboratory. The supernatants of water samples were passed through 0.7 μm GF/F filters (Whatman, UK) using low-pressure suction, then filtered with 0.22 μm and 0.45 μm filters and stored at 4 °C in the dark for short-term determination. Figure 1 shows the locations of the sampling sites.

### 2.2. Water Quality Analysis

The total phosphorus (TP) concentrations of samples were determined using ammonium molybdate spectrophotometry (UV–VisTU1901, Shimadzu, Japan) and total nitrogen (TN) concentrations of samples were determined using potassium persulfate oxidation-ultraviolet spectrophotometry (UV–VisTU1901, Shimadzu, Japan). The chemical oxygen demand COD concentrations of the samples were determined using the potassium dichromate method and dissolved organic carbon (DOC) concentrations of samples were determined using a TOC (total organic carbon) analyzer (TOC-LCPNCN200, Shimadzu, Japan). The basic item standard limits of surface water environmental quality standards shown in Table 1 [29].

### 2.3. Fluorescence Properties and Component Analysis of DOM

The UV–Vis was measured in water samples filtered through a 0.45 μm filter using a spectrophotometer (UV–Vis 2550, Shimadzu, Japan). The absorption spectrum was scanned in the wavelength range of 200~800 nm with a 5 cm optical path quartz cuvette at a scanning interval of 1 nm. Ultrapure water was used as a blank, and the absorbance at 700 nm was subtracted to eliminate any scattering effect of particles in the filtrate. The absorption coefficient (a(λ)) was obtained by multiplying by 2.303/r, where r is the length (m) of the optical path of the cuvette. The UV–Vis spectral indices a_280_ and a_355_ can accurately determine the concentrations of DOM in water. We calculated the Napierian absorption coefficients at 280 and 355 nm m^−1^. The specific ultraviolet absorption coefficient at 254 nm (SUVA_254_) was calculated by dividing the decadic absorption coefficient at 254 (in m^−1^) by the concentration of DOC (in mg L^−1^) [30,31].

The 3D fluorescence spectra of water samples were analyzed using a fluorescence spectrophotometer (F-7000, Hitachi, Japan). The measurement conditions were as follows: the excitation light source was a 150 W xenon arc lamp, the negative pressure of the radio multiplier was set to 700 V, the signal-to-noise ratio was greater than 110, the bandpass was λ_Ex_ = 5 nm, λ_Em_ = 5 nm, the response time was selected automatically, and the scanning speed was 2400 nm·min^−1^. The excitation wavelengths (λ_Ex_) were in the range of 200~450 nm, and the scanning interval was 5 nm. The emission wavelengths (λ_Em_) were in the range of 250~600 nm, and the scanning interval was 1 nm. Milli-Q ultrapure water was used as the experimental blank. The Raman spectral intensity of Milli-Q ultrapure water was examined every 15 water samples to monitor the stability of the fluorometer. Raman scattering and Rayleigh scattering were accounted for by subtracting blank water samples and manually setting zero.

Fluorescence index (FI) refers to the ratio of fluorescence intensity at 450 nm and 500 nm when the excitation wavelength is 370 nm. It is used to distinguish the relative contribution of DOM from terrestrial and biological sources [32]. The biological index (BIX) was the ratio of fluorescence intensity at 380 nm and 430 nm when the excitation wavelength was 310 nm [33]. Humification index (HIX) was the integral value of the fluorescence peak between 435~480 nm and 300~345 nm at 254 nm laser wavelength. The higher the HIX value, the higher the humification degree of DOM [34]. Fn (280) refers to the maximum fluorescence intensity between 340 to 360 nm when Ex = 280 nm and represents the relative concentration of protein-like substances [35]. Fn (355) refers to the maximum fluorescence intensity between 440~470 nm when Ex = 355 nm, which represents the relative concentration of humic-like substances, and the contribution of autobiogenic or terrigenous sources to DOM components [36].

By calculating the FI, HIX, and BIX of DOM in the overlying water, the sources and properties of DOM were determined. All EEM fluorescence data were modeled by the PARAFAC method using the DOM Fluor toolbox in MATLAB R2016b (Mathworks, Natick, MA, USA), the detailed steps can be found in the method used by Wang et al. [37]. Fluorescence components were obtained based on half-split validation and random initialization. Their relative abundances (%) and ratios were determined based on maximum fluorescence intensities (F_max_). The principal components were extracted from the water quality factors related and DOM components using principal component analysis (PCA).The analysis of fluorescence spectrum parameters were shown in Table 2. The excitation and emission maxima of the fluorescent components of DOM identified in the overlying water by parallel factor (PARAFAC) modeling were shown in in Table 3.

## 3. Results

### 3.1. Water Quality Features of the Wanggang River

Temporal and spatial distributions of water quality features of the Wanggang River were shown in Figure 2. Pollutant concentrations in the Wanggang River during the dry season were higher than during the wet season. Temporally, the average TP concentration during the dry season was 0.15 mg/L, which was significantly higher than during the wet season (*p* < 0.01). Both dry and wet seasons met the surface water quality standard of class Ⅲ. The average COD concentration during the dry season was 28.00 mg/L, which was 2.3 times higher than that during the wet season (12.10 mg/L). On the whole, the wet season met surface water quality standard Ⅲ, and the dry season met surface water quality standard Ⅳ. The TN concentrations during the dry (2.07 mg/L) and wet seasons (1.94 mg/L) were not significantly different. On the whole, both dry and wet seasons met the water quality standard. Spatially, the average TN and TP concentrations in the lower reaches were 2.28 mg/L and 0.15 mg/L, respectively, both of which were higher than those in the upper and middle reaches. The average COD concentration in the lower reaches was 22.27 mg/L, which was 19% higher than in the upper reaches. 

### 3.2. UV–Vis Spectra of DOM

The DOM content in the water was characterized by DOC concentration. The average DOC concentration in the Wanggang River during the dry season was 6.92 mg/L, which was significantly higher than in the wet season (*p* < 0.01), both values were less than the primary discharge standard (20 mg/L). Overall, DOC concentration in the lower reaches was higher than in the other river sections. Figure 3b shows that SUVA_254_ in the water during the wet season was significantly higher; SUVA_254_ in the middle reaches of the Wanggang River was significantly higher than those in the upper and lower reaches during both wet and dry seasons (*p* < 0.01). In this study, a_280_ in water during the dry season was significantly higher than during the wet season (*p* < 0.05). Additionally, the average a_355_ in water during the dry season was 5.38, which was significantly higher than that during the wet season (*p* < 0.05).

### 3.3. Fluorescence Spectra of DOM

#### 3.3.1. Fluorescence Spectral Parameters of DOM

Figure 4 illustrates the variability in the fluorescence spectral parameters of the DOM in the Wanggang River. In terms of time, the FI of the Wanggang River was relatively high during the wet season (up to 3.43), and the FI values in both dry and wet seasons were greater than 1.9. The BIX showed no significant differences between the dry and wet seasons and was generally lower than 1, ranging between 0.8 and 1.0. Fn (280) during the dry season was significantly higher than that during the wet season (3.4 times; *p* < 0.01). Spatially, the FI of the lower reaches (2.64) was higher than the upper and middle reaches. The spatial distributions of HIX and Fn (280) were significantly different between the dry and wet seasons. The HIX of the middle reaches peaked (~1.47) during the dry season (HIX < 1.5); while the HIX of the upper reaches peaked (~3.66) during the wet season (HIX < 6). The Fn (280) of the middle reaches was lowest during the dry season and peaked in the upstream reaches during the wet season.

#### 3.3.2. Fluorescence Components of DOM in Water

The 3D fluorescence data of DOM in the Wanggang River were analyzed using the PARAFAC model, and the three largest fluorescence components were obtained via co-analysis (Figure 5). Two protein-like components (C1, C2) and one humic-like component (C3) were obtained during both the dry and wet seasons. The emission peak positions of C1 and C2 were both located at the emission wavelength Em < 380 nm. C1 (λ_Ex_/λ_Em_ = 235/280 nm, 350 nm) was indicative of tryptophan protein-like substance peaks T1 and T2, which typically originate from the degradation by microbes and can indirectly reflect the activity of microbes in water; C2 (λ_Ex_/λ_Em_ = 270 nm/320 nm) indicated tyrosine protein peak B, which is generally considered to be produced mainly by microbial degradation. C3 (λ_Ex_/λ_Em_ = 260 nm/434 nm) was located at emission wavelength Em > 380 nm, containing one excitation peak and one emission peak, indicative of humic-like peak A in the UV region. Peak A was related to the high aromaticity and high molecular weight groups in DOM, which are not easily biodegraded and utilized, and could indicate exogenous inputs.

Figure 6 shows the relative ratios of the fluorescent components of DOM in the Wanggang River. As observed, the percentages of the total DOM fluorescence made up of protein-like components during the dry and wet seasons were 80.38% and 60.04%, respectively. Tryptophan protein-like component C1 made up 45.60% and 40.67% of the total fluorescence intensity during the dry and wet seasons, and tyrosine protein component C2 made up 34.78% and 19.37% of the total fluorescence intensity during the dry and wet seasons. Humic-like component C3 made up 19.63% and 39.96% of the UV fluorescence intensity during the dry and wet seasons, respectively.

## 4. Discussion

### 4.1. Composition and Distribution Characteristics of DOM in the Wanggang River

Overall, the fluorescence intensity of DOM in the Wanggang River was high during the dry season and low in the wet season; it also increased from the upper to lower reaches. The DOM in the Wanggang River mainly consisted of protein-like substances (mainly tryptophan-like components) with the contents of protein-like components C1 and C2 exceeding 60% (Figure 6). Fn (280) was significantly higher during the dry season than in the wet season (Figure 4), suggesting high relative concentrations of protein-like substances during the dry season. C3 content was significantly higher during the dry season than the wet season and increased in variability from the upper to lower reaches (Figure 6). This was mainly because the land in the lower reaches was mostly used for farming, which increased the input of humic-like components into the lower reaches of the Wanggang River via surface runoff [41]. The SUVA_254_ during the wet season was significantly higher than during the dry season, suggesting that the composition of aromatic species in the aquatic DOM was higher during the wet season, which was consistent with He [42]. In this study, a_280_ and a_355_ in water were significantly higher during the dry season than wet season (Figure 3c,d), and a_280_ during the dry season was significantly higher than a_355_, suggesting that the concentrations of protein-like components of DOM in the Wanggang River were higher than humic-like components during the dry season. This was mainly due to the smaller volume of river water during the dry season, which was strongly influenced by the large amount of DOM from the degradation of phytoplankton and microbes in the river and the influx of DOM in tributaries which were strongly influenced by external sources [43].

### 4.2. Source Identification of DOM in the Wanggang River

The DOM in the Wanggang River during the dry and wet seasons was mainly endogenous. The FI values of DOM in the Wanggang River were all greater than 1.9, suggesting that 241 microbial metabolism was the dominant source of DOM in the water [44,45]. The BIX ranged between 0.8 and 1.0, suggesting that the DOM in the Wanggang River had more endogenous neogenesis, and was mainly autogenic, which is consistent with Zhang et.al. [25]. The HIX values indicated that the microbial sources of DOM in water were stronger during the dry season. During the wet season, the DOM spectra of the upper and middle reaches had stronger humic signals than the lower reaches, their characteristics indicating that newly generated autogenic sources were also stronger [46,47]. The tryptophan and tyrosine components were mainly sourced from microbial degradation in water, that is, the protein-like components from the degradation of phytoplankton in rivers [48,49]. C1 showed a significant positive correlation with C2 and C3 (*p* < 0.01), suggesting that the three components might be homologous. During the dry season, fluorescence intensities in the different river sections were ordered: lower reaches > upper reaches > middle reaches. This distribution was mainly caused by the large number of industrial enterprises and aquaculture facilities in the middle reaches, which discharge pollutants into the Wanggang River. In most cases, the microbial degradation of DOM was high, and C1 and C2 contents were low. Hence, the DOM in the middle reaches of the Wanggang River exhibited a high degree of biodegradation, which was consistent with the fluorescence intensity results. Additionally, the underwater light field in the wet and dry seasons will also change, prompting the photodegradation of DOM [25]. Overall, the DOM in water exhibited obvious autogenic characteristics during the dry season. The fluorescence intensity of DOM in water during the wet season was generally weaker than in the dry season, following the order: lower reaches > middle reaches > upper reaches, and the lower reaches had larger contributions from biogenic sources.

### 4.3. Factors Influencing DOM in the Wanggang River

The initial eigenvalues of the four principal components (the components C1, HIX, TN, and FI) were all greater than 1 (Figure 7), and their cumulative contribution to the variance was 78.69%, indicating that they contain most of the characteristics of the original indicators. The contribution of PC1 was 37.60%, and C1 was the most descriptive indicator of this component, therefore PC1 was related to the content of the tryptophan-like components of DOM in water. The contribution of PC2 was 18.44%, and HIX was the descriptive indicator for this component indicator. Therefore, PC2 was related to the weak humic characteristics of DOM in water. The contribution of PC3 was 12.62%, and TN was the descriptive indicator of this component, suggesting that PC3 was related to N content in water, that is, the contributions of domestic sewage and farmland fertilization to DOM in water. The contribution of PC4 was 10.03%, and FI was the descriptive indicator for this component, reflecting the contribution of autogenic characteristics to DOM in water. The correlations between fluorescent components and the water-quality parameters of DOM in the Wanggang River were further compared (Figure 7). TP was significantly positively correlated with C1 and C2 (*p* < 0.01). COD was positively correlated with C1 and C2 (*p* < 0.05), suggesting that the concentration, migration, and transformation of phosphorus-containing compounds and organic pollutants affected the protein-like components of DOM in the Wanggang River [50,51]. The HIX was significantly negatively correlated with C1 and C2 (*p* < 0.01), suggesting that the content of humic-like components affected the concentrations of C1 and C2, and was negatively correlated with protein-like content. C3 had a significant positive correlation with a_280_ and a_355_ (*p* < 0.01), suggesting that C3 might have dual sources, products of terrestrial humic and microbial degradation and metabolism, i.e., autogenous sources of DOM had some influence on its content.

## 5. Conclusions

The compositions and sources of DOM in the Wanggang River during the dry and wet seasons were investigated. The results demonstrated that the DOM content in the Wanggang River during the dry season was significantly higher than during the wet season. The DOM content in the Wanggang River increased from the upper to lower reaches. The most important principal components of DOM in the Wanggang River primarily reflected protein-like components. The relative content of protein-like components in the Wanggang River during the dry season was higher than that in the wet season, but the content of humic-like components in the Wanggang River fluctuates from upstream to downstream. The DOM in the Wanggang River was mainly endogenous, mainly sourced from the biodegradation of phytoplankton in water. The DOM during the dry season had more obvious autogenic characteristics compared to the wet season. This study traced the source of pollutants by analyzing the composition characteristics of DOM, which provides new insights into biogeochemical cycling processes and offers a theoretical basis for better management of rivers flowing into the sea.

## Figures and Tables

**Figure 1 ijerph-19-09219-f001:**
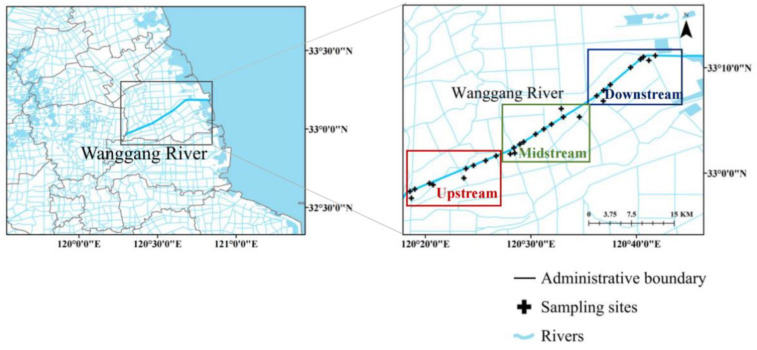
Map of sampling sites.

**Figure 2 ijerph-19-09219-f002:**
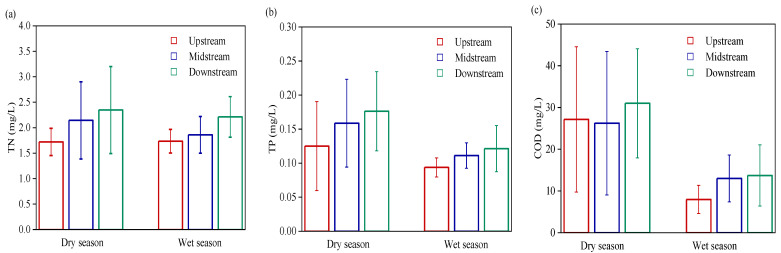
Temporal and spatial distributions of water quality features of the Wanggang River. (**a**) TN; (**b**) TP; (**c**) COD.

**Figure 3 ijerph-19-09219-f003:**
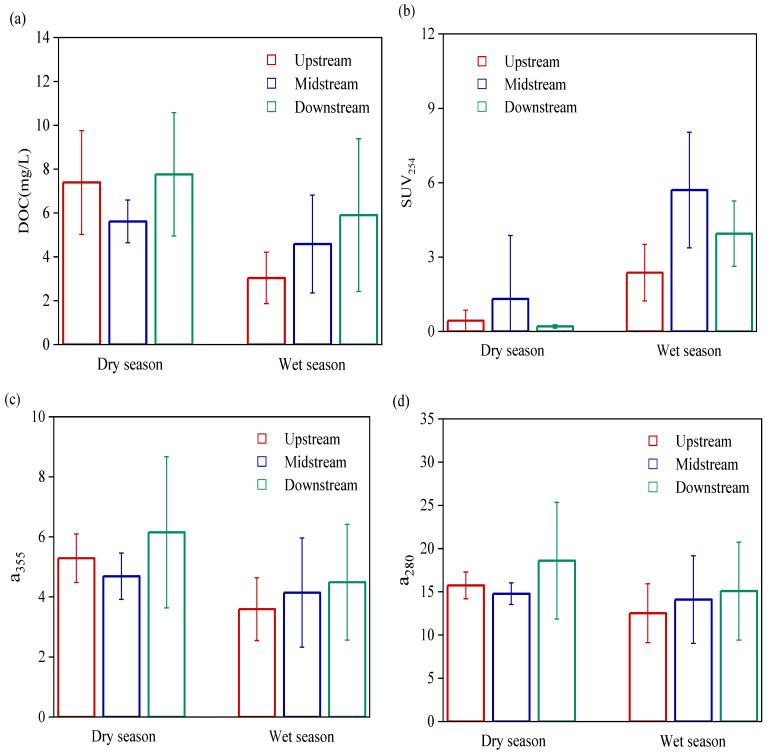
Distributions of DOC and UV–Vis spectral parameters in the Wanggang River. (**a**) DOC; (**b**) SUV_254_; (**c**) a_355_; (**d**) a_280_.

**Figure 4 ijerph-19-09219-f004:**
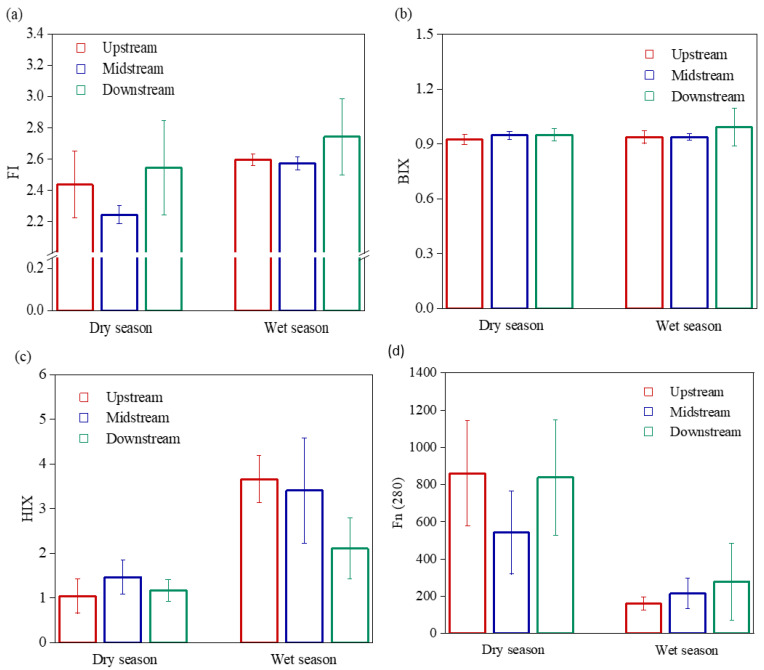
Fluorescence spectral parameters of DOM in the Wanggang River. (**a**) FI; (**b**) BIX; (**c**) HIX; (**d**) Fn(280).

**Figure 5 ijerph-19-09219-f005:**
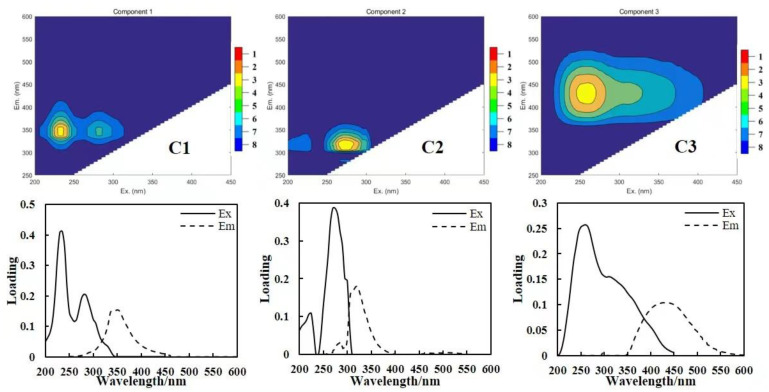
Excitation emission fluorescence spectra obtained by PARAFAC.

**Figure 6 ijerph-19-09219-f006:**
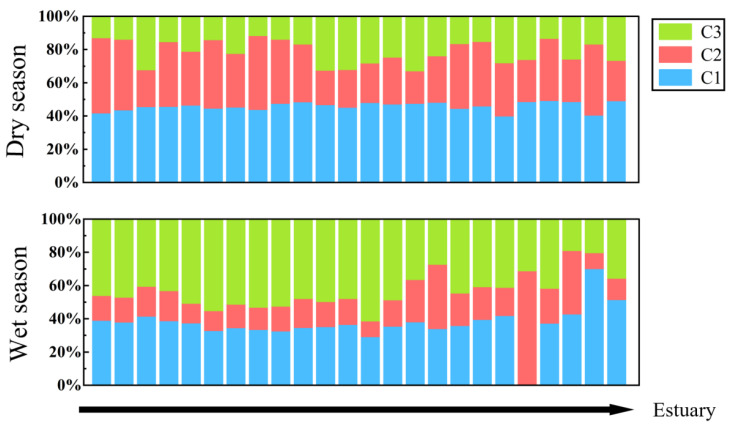
Relative ratios of fluorescence components of DOM in the Wanggang River.

**Figure 7 ijerph-19-09219-f007:**
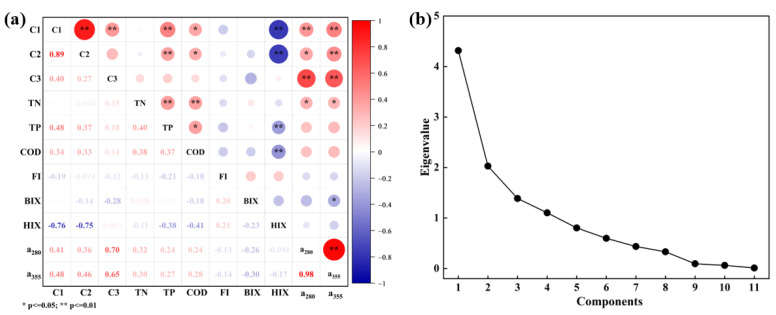
(**a**) Correlation analysis of DOM components and various water−quality indices in the Wanggang River; (**b**) Eigenvalues of principal components of DOM components and water-quality indices in the Wanggang River.

**Table 1 ijerph-19-09219-t001:** Basic item standard limits of surface water environmental quality standard. mg/L.

		I	II	III	IV	V
1	TP	≤0.02	≤0.1	≤0.2	≤0.3	≤0.4
2	TN	≤0.2	≤0.5	≤1.0	≤1.5	≤2.0
3	COD	≤15	≤15	≤20	≤30	≤40

**Table 2 ijerph-19-09219-t002:** Analysis of fluorescence spectrum parameters [32,33,34,35,36].

Component	Value Range	Characterization Results
FI	FI > 1.9	mainly derive from the microbial activity of water body with obvious autobiogenic characteristics
BIX	0.8 < BIX < 1.0	represents that there are many endogenous DOM, and most of them are autogenic
HIX	<1.5	represents biological or aquatic bacterial sources
	1.5 < HIX < 3	represents weak humus and important recent authigenic characteristics
	3 < HIX < 6	represents strong humic and weak recent authigenic characteristics
Fn (280)	-	represents the relative concentration of protein-like substances
Fn (355)	-	represents the relative concentration of humic-like substances

**Table 3 ijerph-19-09219-t003:** Excitation and emission maxima of the fluorescent components of DOM identified in the overlying water by parallel factor (PARAFAC) modeling [38,39,40].

Component	Ex_max_ (nm)	Em_max_ (nm)	Description
C1	235/280	350	tryptophan protein-like substance peaks T1 and T2
C2	270	320	tyrosine protein peak B
C3	260	434	humic-like peak A in the ultraviolet region

## Data Availability

Not applicable.

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
