# Peer review of "The Distribution of DOM in the Wanggang River Flowing into the East China Sea"

_ijerph, 2022, doi:10.3390/ijerph19159219_

Round 1

Reviewer 1 Report

The content of the paper entitled "Hydrological conditions controlling the distribution of DOM in the Wanggang river flowing into the East China Sea" is not in the line with the title of the paper. The focus of the study is on the DOM components, rather than hydrological conditions. It is not analyzed any hydrological conditions (such as water level and flow) and their impact on the distribution of DOM. The authors only mentioned differences in DOM distribution between wet and dry seasons, as well as in upper and low reaches. The focus of the paper should be on the hydrological conditions throughout the whole paper. If it is not possible, the authors should change the title of the paper (suggestion: The distribution of DOM in the Wanggang river flowing into the East China Sea”.

Authors should provide a short description of the study area.

Section Material and Methods should be better presented. Authors should describe (in short) applied methods (such as water quality indices, spectral indices, PARAFAC model, etc.) with a focus on standard values and ranges of indicators.

Authors provide only numbers as results without explanation. It is necessary to compare obtained results with the standards and discuss them in this manner. Also, the authors should compare their results with some previous investigations.

The conclusion should be expanded, with the focus on the hydrologic conditions.

Some references do not support the text and should be deleted and/or replaced with appropriate ones.

Some Figures are missing (only their titles are given).

Other comments are given in the manuscript body.

Author Response

Response to Reviewer 1 Comments

Thank you for your letter and for the reviewers’ comments concerning our manuscript entitled “Hydrological conditions controlling the distribution of DOM in the Wanggang river flowing into the East China Sea”. Those comments are all valuable and very helpful for revising and improving our paper, as well as the important guiding significance to our research. Taking account of reviewers’ comments, we have revised and improved the manuscript. We hope our revisions meet with approval. Revised portion is marked with blue in the paper. The main corrections in the paper and the responses to the reviewers’ comments are as follows

Point 1: The content of the paper entitled "Hydrological conditions controlling the distribution of DOM in the Wanggang river flowing into the East China Sea" is not in the line with the title of the paper. The focus of the study is on the DOM components, rather than hydrological conditions. It is not analyzed any hydrological conditions (such as water level and flow) and their impact on the distribution of DOM. The authors only mentioned differences in DOM distribution between wet and dry seasons, as well as in upper and low reaches. The focus of the paper should be on the hydrological conditions throughout the whole paper. If it is not possible, the authors should change the title of the paper (suggestion: The distribution of DOM in the Wanggang river flowing into the East China Sea”.

Reply: Thanks for the reviewer’s suggestion. We have rephrased the title of the paper. At the same time, some content modifications were made in the revised version.

Point 2: Authors should provide a short description of the study area.

Reply:Thanks for the reviewer’s suggestion. We have provide a short description of the study area on Line 65-67 of the revised version.

Point 3: Section Material and Methods should be better presented. Authors should describe (in short) applied methods (such as water quality indices, spectral indices, PARAFAC model, etc.) with a focus on standard values and ranges of indicators.

Reply: Thanks for the reviewer’s suggestion. We have provided the standard limits of surface water environmental quality standards in the revised version.

Point 4: Authors provide only numbers as results without explanation. It is necessary to compare obtained results with the standards and discuss them in this manner. Also, the authors should compare their results with some previous investigations.

Reply: Thanks for the reviewer’s suggestion. We have rephrased this sentence in the revised version.

Point 5: Some references do not support the text and should be deleted and/or replaced with appropriate ones.

Reply: Thanks for the reviewer’s suggestion. We have rephrased this references in the revised version. The more detailed explanation attached below.

Point 6: Some Figures are missing (only their titles are given).

Reply: Please accept our apologies for these grammatical mistakes. We have provided the Figures in the revised.

Other comments are given in the manuscript body are as follows:

Point 7: According to reference 4 DOM includes humid acids and fulvic acids. Please cite adequate reference for other highlighted DOM components.

Reply: Thanks for the reviewer’s suggestion. We have cited the adequate reference in the revised version.

[5] Niloy, N M, Haque, Md. Morshedul,and Tareq, Shafi M.. Temporal changes in hydrochemistry and DOM characteristics of the Brahmaputra River: implication to the seasonality of water quality. Environmental Science and Pollution Research . 2022. 29(23):35165-35178. https://doi.org/10.1007/S11356-022-18618-Z.

[6] Fellman, J.B., Hood, E., Spencer, R.G.M. Fluorescence spectroscopy opens new windows into dissolved organic matter dynamics in freshwater ecosystems: a review. Limnol. Oceanogr. 2010. 55: 2452–2462. https://doi.org/10.4319/lo.2010.55.6.2452.

[7] Lee, M.-H., Osburn, C.L., Shin, K.-H., et al. New insight into the applicability of spectroscopic indices for dissolved organic matter (DOM) source discrimination in aquatic systems affected by biogeochemical processes. Water Res. 2018. 147, 164–176. https://doi.org/10.1016/j.watres.2018.09.048.

Point 8: Reference 5 is not adequate in this context. Please replace it with another one or delete

Reply: Thanks for the reviewer’s suggestion. We have deleted this reference in the revised version.

Point 9: This is not clear enough. Central what? What is the central of (or in) the geochemical cycles? Please clarify

Reply: Thanks for the reviewer’s suggestion. We have rephrased this sentence on Line 37 of the revised version.

Point 10: There is no word about marine and terrestrial carbon pools in reference 8. Please provide adequate reference.

Reply: Thanks for the reviewer’s suggestion. We have cited the adequate reference in the revised version.

[10] Pan Y, Birdsey R A, Fang J, et al. A large and persistent carbon sink in the world’s forests. Science, 2011. 333(6045): 988-993. https://doi.org/10.1126/science.1201609.

[11] Ghosh P K, Mahanta S K. Carbon sequestration in the grassland systems. Range Management and Agroforestry, 2014. 35(2): 173-181.

[12] Arrigo K P. Marine manipulations[J].Nature, 2007. 450(7169):491-492. https://doi.org/10.1038/450491a.

Point 11: Is it terrigenous organic matter? Please clarify the highlighted term.

Reply: Thanks for the reviewer’s suggestion. We have rephrased this sentence on Line 43 of the revised version.

Point 12: The reference 13 provides only information about impacts of wastewater. Please provide additional reference(s) to give information impacts of other mentioned activities.

Reply: Thanks for the reviewer’s suggestion. We have cited the adequate reference in the revised version.

[18] Mendonça R, Kosten S, Sobek S, Barros N, Cole J J, Tranvik L, Roland F. Hydroelectric carbon sequestration[J]. Nature Geoscience, 2012. 5(12): 838-840. 18.https://doi.org/10.1038/ngeo1705.

[19] Skalak K J, Benthem A J, Schenk E R, Hupp C R, Galloway J M, Nustad R A. Wiche G J. Large dams and alluvial rivers in the Anthropocene: the impacts of  the Garrison and Oahe Dams on the Upper Missouri River[J]. Anthropocene, 2013. 2: 51-64. https://doi.org/10.1016/j.ancene.2013.10.002.

[20]Amaral V, Romera-Castillo C, García-Delgado M, et al. Distribution of dissolved organic matter in estuaries of the southern Iberian Atlantic Basin: Sources, behavior and export to the coastal zone[J]. Marine Chemistry, 2020. 226, doi:10.1016/j.marchem. 2020. 103857.

Point 13: The reference 14 is not appropriate.

Reply: Thanks for the reviewer’s suggestion. We have cited the adequate reference in the revised version.

[21] Aiken G R, Hsu-Kim H, Ryan J N. Influence of dissolved organic matter on the environmental fate of metals, nanoparticles, and colloids[J].  Environmental Science & Technology, 2011, 45(8):3196-3201. https://doi.org/ 10.1021/es103992s.

[22] Stedmon C A, Markager S, Bro R. Tracing dissolved organic matter in aquatic environments using a new approach to fluorescence spectroscopy[J]. Marine Chemistry, 2003, 82(3-4): 239-254. https://doi.org/10.1016/s0 304-4203(03)00072-0.

[23] Steinberg C. Ecology of humic substances in freshwaters: determinants from geochemistry to ecological niches[M]. Springer Science & Business Media, 2003. ISBN:978-3-642-07873-6.

Point 14: Biological index (BIX) and fluorescence index (FI) were not subject of study in reference 17. Please provide adequate reference

Biological index (BIX) is not subject of the study in reference 18. Please provide additional reference for BIX.

Reply:

Thanks for the reviewer’s suggestion. We have cited the adequate reference in the revised version.

[27] Wickland K P, Neff J C, Aiken G R. Dissolved Organic Carbon in Alaskan Boreal Forest: Sources, Chemical Characteristics, and Biodegradability[J]. Ecosystems. 2007, 10(8): 1323-1340. https://doi.org/10.1007/s 10021-007- 9101-4.

[28] Wilson H F, Xenopoulos M A. Effects of agricultural land use on the composition of fluvial dissolved organic matter[J]. Nature Geoscience. 2009, 2(1): 37-41. 28.https://doi.org/10.1038/ngeo391.

Point 15: Left map should be improved. Please add Scale bar and North arrow. Also, text is not visible.

Reply: Thanks for the reviewer’s suggestion. We have provided a clearer Figure in the revised version.

Point 16: This section should also include Table with standards (classification, ranges) of water quality parameters values.

Reply: Thanks for the reviewer’s suggestion. We have provided the standard limits of surface water environmental quality standards in the revised version.

Point 17: Neither Napierian absorption coefficient nor the C-specific ultraviolet absorption coefficient were not calculated in reference 13. Please provide adequate reference.

Reply: Thanks for the reviewer’s suggestion. We have cited the adequate reference in the revised version.

[31] Singh S, D’Sa E J, Swenson E M. Chromophoric dissolved organic matter (CDOM) variability in Barataria  Basin  using excitation–emission  matrix  (EEM)  fluorescence  and  parallel  factor  analysis (PARAFAC)[J]. Science of the Total Environment, 2010, 408(16): 3211–3222. 31. https://doi.org/10.1016/j. scitotenv. 2010.03.044.

[32] Nishorganic carbon produced by ozonation on biological activated carbon[J]. Chemosphere, 2004,56(2):113-119. https://doi.org/10.1016/j.chemosphere.2004.03.009.

Point 18: Please provide short description about these indices (how are they calculated, which parameters are included, value ranges, etc...)

Reply: Thanks for the reviewer’s suggestion. We have added relevant content in revised version.

Point 19: Here is presented only numbers which does not provide any information about water quality. Water quality parameters should be analysed comparing with water quality parameters standard values.

Reply: Thanks for the reviewer’s suggestion. We have added relevant content in revised version.

Point 20: Figure 2. is not presented in the manuscript. Where is the Figure 2?

Reply:

Please accept our apologies for these grammatical mistakes. We have provided the Figures in the revised.

Point 21 DOC concentration should be also analysed comparing with DOC standard values.

Indices (FI, BIX and HIX) should be analysed comparing with standard values (classification, ranges). Fn should be introduces and explained in the Methods.

Reply: Thanks for the reviewer’s suggestion. We have added relevant content in the revised version.

Point 22 Figure 4 is not presented in the manuscript. Where is the Figure 4?

Reply:Thanks for the reviewer’s suggestion. We have presented Figure 4 in the revised.

Point 23: PARAFAC model should be described in the section Materials and Methods.

Reply: Thanks for the reviewer’s suggestion. We have described it in the section Material and methods.

Point 24: humic or humus. Use one term throughout the whole text

Reply: Please accept our apologies for these grammatical mistakes. We have provided it in the revised.

Point 25: Figure 5 and Figure 6 should be separately presented

Reply: Thanks for the reviewer’s suggestion. We have presented Figure 5 and Figure 6  separately

Point 26: This is not confirmed in reference 13, there is no word about differences of the aromatic species between wet and dry season. Please provide an adequate reference.

Reply: Thanks for the reviewer’s suggestion. We have cited the adequate reference in the revised version.

[39] He D, Wang K, Pang Y, et al. Effects of regulation and storage of the Three Gorges Reservoir on the spectrum and molecular composition of soluble organic matter (DOM) in the main and tributaries of the reservoir. Chinese Society of Mineralogy, Petrogeochemistry: Chinese Society of Mineralogy, petrogeochemistry. 2019. 1.

Point 27: Spectral indices should be explained in the section Materials and methods. This sentence should be moved in the section Material and methods.

Reply: Thanks for the reviewer’s suggestion. We have rephrased it in the section Material and methods in the revised version.

Point 28: Please provide adequate reference. Reference 25 does not support this.

Reply: Thanks for the reviewer’s suggestion. We have cited the adequate reference in the revised version.

[45] Yao X, Zhang Y L, Zhu G W, et al. Resolving the variability of CDOM fluorescence to differentiate the sources and fate of DOM in Lake Taihu and its tributaries [J]. Chemosphere, 2011, 82(2):145-155.https://doi.org /10.1016/j.chemosphere.2010.10.049.

[46] KamJunke N, Ttimpling WV, Hertkom N, et al. A new approach for evaluating transformations of dissolved organic matter(DOM)via high-resolution mass spectrometry and relating it to bacterial activity[J]. Water Research,2017,123:5 13-523. https://doi.org/10.1016/j.watres.2017.07.008.

Point 29: Conclusions should be expanded with the focus on hydrological conditions of DOM distribution.

Reply: Thanks for the reviewer’s suggestion. We have added relevant content in the revised version. we also rephrased the title of this paper.

Reviewer 2 Report

This paper studies the dissolved organic matter distributed along the Wanggang River. Three parts are considered in the Yancheng City, upstream, midstream and downstream, where the water is analysed.

Moreover, wet and dry situations are used. The paper is focused on a noticeable subject, such as the water quality:

However, some minor changes should be introduced previously to its publication in the International Journal of Environmental Research and Public Health.

Measurements were taken in May 2021 and September 2021. Could the authors indicate the dry and wet seasons and the meteorological contrast between them? How many water samples were used?

Another remarks.

Line 35. Suppress the point.

Line 49. “activities direct” or “direct activities”.

Lines 60. “analyzed emission” or “analyzed by emission”

Figure 1. “Upstream” should be “downstream” and vice versa.

Line 85-90. TP, TN, COD, DOC and TOC should be introduced.

Line 130. Revise the quantities.

Line 144. The figure caption should be under the figure.

Line 156. Remove “During the dry season,”.

Line 157-158. Revise the sentence, since the value was the highest for the lower reaches during the wet season.

Figure 6. The colour scale is necessary.

Section 4.3. Change “principle components” by “principal components”.

Author Response

Response to Reviewer 2 Comments

Thank you for your letter and for the reviewers’ comments concerning our manuscript entitled “Hydrological conditions controlling the distribution of DOM in the Wanggang river flowing into the East China Sea”. Those comments are all valuable and very helpful for revising and improving our paper, as well as the important guiding significance to our research. Taking account of reviewers’ comments, we have revised and improved the manuscript. We hope our revisions meet with approval. Revised portion is marked with blue in the paper. The main corrections in the paper and the responses to the reviewers’ comments are as follows

Point 1: Measurements were taken in May 2021 and September 2021. Could the authors indicate the dry and wet seasons and the meteorological contrast between them? How many water samples were used?

Reply: Thanks for the reviewer’s suggestion. We have added some description in the section Material and methods of the revised version.

Another remarks

Point 2: Line 35. Suppress the point.

Reply:Thanks for the reviewer’s suggestion. We have rephrased this sentence on line 37 of the revised version. 

Point 3: Line 49. “activities direct” or “direct activities”.

Reply: Thanks for the reviewer’s suggestion. We have rephrased this sentence in the revised version. 

Point 4: Lines 60. “analyzed emission” or “analyzed by emission”

Reply:Thanks for the reviewer’s suggestion. We have rephrased this sentence in the revised version.

Point 5: Figure 1. “Upstream” should be “downstream” and vice versa.

Reply: Please accept our apologies for this mistake. We have adjusted the description in the revised. 

Point 6: Line 85-90. TP, TN, COD, DOC and TOC should be introduced.

Reply: Really thanks for the reviewer’s kind remind. We have adjusted the description on line 91-99 of the revised version. 

Point 7: Line 130. Revise the quantities.

Reply: Really thanks for the reviewer’s kind remind. The concentration here is the average quantity of wet season and dry season.

Point 8: Line 144. The figure caption should be under the figure.

Reply: Really thanks for the reviewer’s kind remind. We have adjusted the description in the revised. 

Point 9: Line 156. Remove “During the dry season,”.

Reply: Thanks for the reviewer’s suggestion. We have removed it in the revised version. 

Point 10: Line 157-158. Revise the sentence, since the value was the highest for the lower reaches during the wet season.

Reply: Really thanks for the reviewer’s kind remind. We have adjusted the description in the revised

Point 11: Figure 6. The colour scale is necessary.

Reply: Really thanks for the reviewer’s kind remind. We have adjusted the description in the revised

Point 12: Section 4.3. Change “principle components” by “principal components”.

Reply: Really thanks for the reviewer’s kind remind. We have adjusted the description in the revised

Round 2

Reviewer 1 Report

Authors made some improvements, however manuscript still need further revision.

The section Material and Methods still need improvements. Authors provide some description of the applied methods, as well as standard values for some water quality parameters, but standard values for DOC concentrations and indices still lack (numbers without standard values and/or ranges could not provide better understanding). Also, authors should provide short description of the PARAFAC model.

Authors should compare their results with previous studies.

Conclusions should be expanded. It could be added some directions for further investigations.

Some references are inappropriate and should be deleted and/or replaces with adequate ones.

Other comments are given in the manuscript body.

Author Response

Response to Reviewer 1 Comments

Thank you for your letter and for the reviewers’ comments concerning our manuscript. Those comments are all valuable and very helpful for revising and improving our paper, as well as the important guiding significance to our research. Taking account of reviewers’ comments, we have revised and improved the manuscript. We hope our revisions meet with approval. Revised portion is marked with red in the paper. The main corrections in the paper and the responses to the reviewers’ comments are as follows:

Point 1: The section Material and Methods still need improvements. Authors provide some description of the applied methods, as well as standard values for some water quality parameters, but standard values for DOC concentrations and indices still lack (numbers without standard values and/or ranges could not provide better understanding). Also, authors should provide short description of the PARAFAC model. Authors should compare their results with previous studies. 

L122.This section still needs improvement. Please provide ranges for applied indices, as it has been done for TP, TN and COD.

L161. DOC concentration should be also analyzed comparing with DOC standard values

Reply: Thanks for the reviewer’s suggestion. We have improved the section Material and Methods even further. We also compared some data with previous studies and standard values.

Point 2: Conclusions should be expanded. It could be added some directions for further investigations./ L291. Conclusions should be expanded. Results could be compared with other similar studies, some implication for future investigation could be presented.

Reply: Thanks for the reviewer’s suggestion. We have expanded the conclusions in the revised version.

Point 3:Some references are inappropriate and should be deleted and/or replaces with adequate ones.

Reply: Thanks for the reviewer’s suggestion. We have rephrased those references in the revised version. The more detailed explanation attached below.

Other comments are given in the manuscript body.

Point 4: L34. According to the reference 4 DOM includes humid acids and fulvic acids. This reference should be moved in the text after (FA)

Reply: Thanks for the reviewer’s suggestion. We have moved the reference 4 in the text after (FA) in the revised version. 

Point 5: L36. It is still not clear. What affects the biogeochemical cycle? The molecular structure, content, aromaticity and degree of humification of DOM? Please clarify.

Reply: Thanks for the reviewer’s suggestion. In order to be more clear, we have deleted this sentence in the revised version.

Point 6: L51.Reference 21 is not adequate in this context. This study is about impact of DOM on other substances, not about impact of human activities on DOM.

Reply: Thanks for the reviewer’s suggestion. We have added adequate references in the revised version.

[21]Wang, Y.L., Hu, Y.Y., Yang, C.M., et al. Variations of DOM quantity and compositions along WWTPs-river-lake continuum: implications for watershed environmental management. Chemosphere, 2019, 218: 468-476.  https://doi.org/10.1016/j.chemosphere.2018.11.037.

Point 7: L56. This is reference 22. Please correct it as well as order of succeeding references

Reply: Please accept our apology for this grammatical mistake. We have improved it in the revised.

Point 8: L60. References 29-30 do not provide information about BIX. Please provide adequate references.

Reply: Thanks for the reviewer’s suggestion. We have added adequate references in the revised version. These two references provide detail information about BIX.

[6]Fellman, J.B., Hood, E., Spencer, R.G.M. Fluorescence spectroscopy opens new windows into dissolved organic matter dynamics in freshwater ecosystems: a review. Limnol. Oceanogr. 2010. 55: 2452–2462. https://doi.org/10.4319/lo.2010.55.6.2452.

[28]Huguet, A., Vacher, L., Relexans, S., Saubusse, S., Froidefond, J.M. and Parlanti, E. Properties of fluorescent dissolved organic matter in the Gironde Estuary. Organic Geochemistry. 2009. 40:706-719. https://doi.org/10.1016/j.orggeochem.2009.03.002.

Point 9: L109. Neither Napierian absorption coefficient nor the C-specific ultraviolet absorption coefficient were not calculated in reference 32.

Reply: Thanks for the reviewer’s suggestion. We have added adequate references in the revised version.

[30] Mann, P.J., Davydova, A., Zimov, N., Spencer, R.G.M., Davydov, S., & Bulygina, E., et al. Controls on the composition and lability of dissolved organic matter in Siberia's Kolyma River basin[J]. Journal of Geophysical Research: Biogeosciences, 2012. 117. https://doi.org/10.1029/2011JG001798.

Point 10: L139. Please rephrase to be clear. Suggestion: "Four principle components including"(list which components).../  L264. list which components

Reply: Thanks for the reviewer’s suggestion. We have listed DOM components in Table 3 in the revised version.

Point 11: L149. 2.3 times higher

Reply: Please accept our apology for this grammatical mistake. We have improved it in the revised.

Point 12: L173. Indices (FI, BIX and HIX) should be analyzed comparing with standard values (classification, ranges).

Reply: Thanks for the reviewer’s suggestion.We have improved it on page 5, Line 184-197 in the revised. We also provided the analysis of fluorescence spectrum parameters in Table 2.

Point 13: L191. PARAFAC model should be described in the section Materials and Methods.

Reply: Thanks for the reviewer’s suggestion. We have rephrased this sentence and in the section Materials and Methods. We also cited the adequate reference in the revised version. The detailed steps of PARAFAC model can be found in the method used by Wang et al. (Wang, Y.L., Hu, Y.Y., Yang, C.M., et al. Variations of DOM quantity and compositions along WWTPs-river-lake continuum: implications for watershed environmental management. Chemosphere, 2019, 218: 468-476.  https://doi.org/10.1016/j.chemosphere. 2018.11.037.)

Point 15: Wrong link in References

Reply: Please accept our apologies for those mistakes. We have rephrased it in the revised.

Round 3

Reviewer 1 Report

Authors have significantly improved the manuscript. Some minor corrections still could be done.

Authors should discuss their findings comparing them with other similar studies.

Some references are still inadequate. Authors should check the links for each reference.

Other comments are given in manuscript body.  

Author Response

Response to Reviewer 1 Comments

Thank you for your letter and for the reviewers’ comments concerning our manuscript. Those comments are all valuable and very helpful for revising and improving our paper, as well as the important guiding significance to our research. Taking account of reviewers’ comments, we have revised and improved the manuscript. We hope our revisions meet with approval. Revised portion is marked with red in the paper. The main corrections in the paper and the responses to the reviewers’ comments are as follows:

Point 1: Authors should discuss their findings comparing them with other similar studies.

Reply: Thanks for the reviewer’s suggestion. We have rephrased those sentences in the revised version.

Point 2: Some references are still inadequate. Authors should check the links for each reference.

Reply: Thanks for the reviewer’s suggestion. We have rephrased those references in the revised version. The more detailed explanation attached below.

Other comments are given in manuscript body.  

Point 3: L28, Hydrological conditions (water level, flow, temperature) are not analyzed, and they should be deleted from keywords.

Reply: Thanks for the reviewer’s suggestion. We have deleted this keyword.

Point 4:L65-70,This part should be in new passage.

Reply: Thanks for the reviewer’s suggestion. We have improved it in the revised.

Point 5:Reference 25 is not adequate. There is no word about biological index (BIX) and fluorescence index (FI). References 26 and 27 do not provide information about BIX. Please provide adequate references.

Reply: Thanks for the reviewer’s suggestion. We have added adequate references in the revised version.

  • Zhang,H., Cui K., Guo Z., et al. Spatiotemporal variations of spectral characteristics of dissolved organic matter in river flowing into a key drinking water source in China. Sci Total Environ, 2019, 700:  https://doi.org/10.1016/j.scitotenv.2019.134360.
  • Jiang T., Kaal J., Liang J., et al. Composition of dissolved organic matter (DOM) from periodically submerged soils in the Three Gorges Reservoir areas as determined by elemental and optical analysis, infrared spectroscopy, pyrolysis-GC-MS and thermally assisted hydrolysis and methylation. Sci Total Environ, 2017, 603-604:461.https://doi.org/10.1016/j.scitotenv.2017.06.114.
  • Wilson H., Xenopoulos M. Effects of agricultural land use on the composition of fluvial dissolved organic matter. Nature Geosci, 2009, 2(1):37-41.https://doi.org/10.1038/ngeo391.

Point 6:35 is not appropriate reference. There is no word about Fn(355). Please provide adequate reference.

Reply: Thanks for the reviewer’s suggestion. We have added adequate reference in the revised version.

[36] Rochelle-Newall E J , Fisher T R . Chromophoric dissolved organic matter and dissolved organic carbon in Chesapeake Bay. Mar Chem, 2002, 77(1):23-41. https://doi.org/10.1016/S0304-4203(01)00073-1.

Point 7:Please add a reference for the Table 1. Who developed this classification?

Reply: Thanks for the reviewer’s suggestion. We have added a reference in the revised version.

[29] Environmental quality standards for surface water in China (GB 3838-2002)

Point 8:References 20、21、22、26、28、33、34 are wrong link. Please correct.

Reply: Thanks for the reviewer’s suggestion. We have rephrased those references in the revised version.

Point 9:Some of journal titles are written in full, while others are written as abbreviations. Please unify it.

Reply: Thanks for the reviewer’s suggestion. We have unified the abbreviations.